# Optimization control of all-terrain rescue lift vehicle safety performance based on state feedback

Jian Yao[1,2]☯, Weiyu Qu[2]☯, Dongmei Tian[2*¤], Jimao Shi[2], Jiayun Wang[2], Baiyou Xu[3], Shouyi Wang[2]

1 School of Resources and Safety Engineering, University of Science and Technology Beijing, Beijing, PR China, 2 School of Safety Engineering, North China Institute of Science and Technology, Beijing, PR China, 3 School of Chemistry and Materials Science, Hebei University, Baoding City, Hebei Province, PR China

☯ These authors contributed equally to this work.
¤ Present address: School of Safety Engineering, North China Institute of Science and Technology, No. 368 North Shimen Road, Shijingshan District, Beijing, China.
* tdm1999@126.com

## Abstract

The all-terrain rescue lift vehicle is instrumental in mining emergency rescue operations, with its operational stability being of utmost importance. This study focuses on the XZJ5240JQZ30 all-terrain rescue lift vehicle, optimizing its vehicle structure and steering system. A linear 2DOF model and a PID gain model were developed based on actual vehicle parameters. A feedback system was employed to adjust the rear-wheel steering angle, enabling four-wheel steering (4WS) vehicle control. Numerical simulations were conducted using TruckSim and Simulink software. Utilizing the classic Double lane-change scenario as a test scenario, the study compared variations in the vehicle's centroid slip angle and yaw rate at different speeds, analyzing the impact of PID gain on steering stability. Moreover, the relationship between the centroid height and 4WS vehicle stability at low speeds was examined. Based on these findings, practical application tests were performed on the XZJ5240JQZ30 all-terrain rescue lift vehicle, obtaining relevant data on steering angle error. The results indicate that vehicles equipped with the PID-optimized control system demonstrate significantly higher steering stability than those without it. Furthermore, in practical applications, the actual steering angle closely aligns with the theoretical values. This demonstrates that the proposed optimized control system has substantial practical application value.

## 1. Introduction

In recent years, the safety level of the coal industry has improved considerably due to advancements in safety management methods and emergency rescue technologies.

**Data availability statement:** All data set files are available from the Zenodo database (Accession number: https://doi.org/10.5281/zenodo.15030634).

**Funding:** This study was supported by the National Natural Science Foundation of China under the Youth Science Foundation Project (No. 51704118), the National Key Research and Development Program of China (No. 2018YFC0808200), the Safety Production Science and Technology Project of the Ministry of Emergency Management (No. zhishu-0013-2016AQ) and the Central Universities Fund Support (Nos. AQ1201A and 3142015105). The funders had no role in study design, data collection and analysis, decision to publish, or preparation of the manuscript.

**Competing interests:** The authors have declared that no competing interests exist.

However, coal mining accidents persist due to the complexity of mining processes and harsh production conditions, resulting in an ongoing challenging safety situation. As coal mining operations extend to greater depths, the complexity and risks associated with emergency rescues have increased significantly. Consequently, enhancing the performance of all-terrain rescue lift vehicles is of critical importance for improving and guiding emergency rescue efforts.

During the Chilean mine rescue operation, scholars proposed a concept of utilizing drilling technology for rescue purposes [1,2]. They analyzed potential emergency scenarios and operational details, establishing a foundation for the development of large-diameter drilling technology. Qian Q et al. [3] introduced a novel rescue lift vehicle with a design that enhanced overall integrity and correlation, improving rescue efficiency. As technology advances, vehicle intelligence has progressively improved. Irshayyid A et al. [4] highlighted that Deep Reinforcement Learning (DRL) presents promising opportunities for training autonomous vehicles to manage complex driving tasks. Spielberg N A et al. [5] introduced a physics model-driven neural network structure for vehicle automation. This model can accurately predict road surface conditions without explicitly calculating road friction, even on varying terrains. To address the demands of modern mining rescue missions, which require higher intelligence levels and enhanced off-road capabilities for rescue equipment, it is essential to develop rescue lift devices with increased intelligence and superior adaptability to harsh road conditions [6,7]. Pilutti T et al. [8] pioneered the use of differential braking to analyze vehicle steering stability. Doumiati M et al. [9] proposed a control system that stabilizes vehicles to achieve desired yaw rates through collaborative steering and braking efforts, demonstrating the effectiveness of the linear matrix inequality (LMI) control scheme in critical driving conditions. Aouadj N et al. [10] enhanced vehicle maneuverability and stability by coordinating active front steering and direct yaw control, ensuring improved steering performance while maintaining stability in extreme driving scenarios. Zheng Z et al. [11] proposed a feedforward + predictive Linear Quadratic Regulator (LQR) lateral motion control algorithm based on Genetic Algorithm (GA) parameter optimization and PID steering angle compensation, enhancing vehicle tracking accuracy and stability. The simulation results of a case study demonstrated that LQR and Model Predictive Control (MPC) could prevent rollover with a relatively rapid response when the forklift carries a half load and moves at a slow speed. Ariff M H M et al. [12] proposed a Four-Wheel Active Steering (4WAS) system applicable to both low-speed and high-speed operations based on optimal control theory. The results showed that, at low speeds, the turning radius was reduced compared to Front-Wheel Steering (FWS) vehicles. Zhang Y et al. [13], based on quadratic optimal control theory, proposed an optimal control approach for a four-wheel steering system and conducted simulations using MATLAB/Simulink. The results indicated that the optimal control-based 4WS system could enhance vehicle slip and yaw stability. A review of existing research reveals that current studies primarily utilize optimization control methods such as PID, LQR, and Robust H-infinity Control [14–16] to improve vehicle steering stability. However, these studies mainly focus on passenger cars and

small construction vehicles [16–18], with limited research on large-scale mining engineering vehicles such as all-terrain rescue lift vehicles.

Mining areas encompass extensive territories with complex and variable road conditions within the pits. During rescue operations, vehicle steering stability is paramount. While advancements in mine road planning have generally eliminated the need for continuous or excessively sharp turns in rescue routes, the substantial mass of all-terrain rescue lift vehicles still poses a significant risk of sideslip and yaw accidents during turning maneuvers. This study aims to enhance the steering stability and road adaptability of these vehicles by investigating the factors affecting the steering stability of large-scale mining engineering vehicles and addressing existing issues. The XZJ5240JQZ30 all-terrain rescue lift vehicle was selected as the research subject, with optimizations made to both its structure and steering system. A linear 2DOF model and a PID gain model were developed for the vehicle. Numerical simulations were conducted using Trucksim and Simulink, employing a classic Double lane-change scenario to analyze the impact of PID gain on vehicle steering stability under varying speeds and centroid heights. Based on these findings, a practical application test was performed on the XZJ5240JQZ30 all-terrain rescue lift vehicle to obtain relevant steering angle error data. This research provides a theoretical foundation for the design and optimization of steering stability in all-terrain rescue lift vehicles.

## 2. Optimization design of the all-terrain rescue lift vehicle structure

### 2.1. Basic parameters of the all-terrain rescue lift vehicle

The XZJ5240JQZ30 all-terrain rescue lift vehicle is extensively utilized in contemporary mining emergency rescue operations. Fig 1 illustrates a schematic diagram of its vehicle structure.

To enhance adaptability to challenging road conditions in mining areas, this study focuses on improving the vehicle's structure and steering system based on its fundamental parameters. These modifications aim to enhance driving performance and steering stability. Table 1 presents the basic parameters of the vehicle.

### 2.2. Chassis design

The XZJ5240JQZ30 all-terrain rescue lift vehicle incorporates the XCA30-JY chassis crane technology platform in its design. The design process involved various optimization techniques for chassis lightweighting, including the implementation of a variable cross-section thin-walled box structure for the frame, connecting thin and thick plates on the upper cover

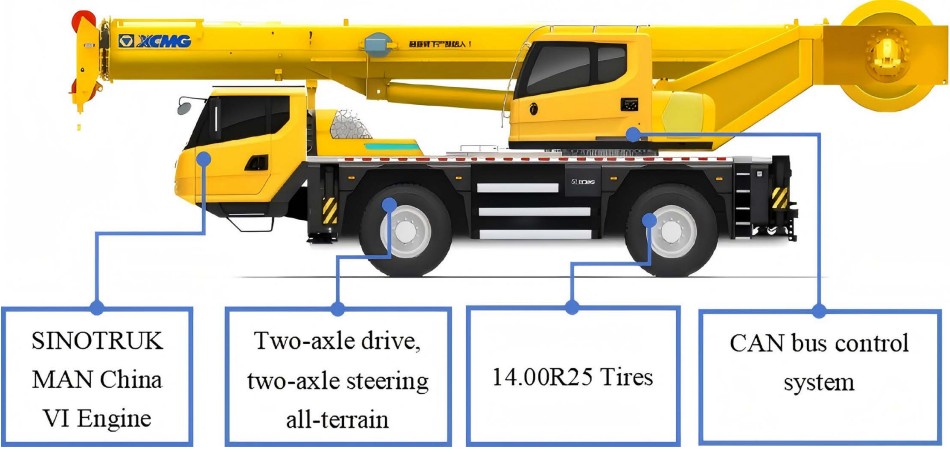

**Fig 1. Schematic diagram of the all-terrain rescue lift vehicle.**

Table 1. Basic parameters of the all-terrain rescue lift vehicle.

| Parameter Name | Parameter Meaning | Parameter Value |
| --- | --- | --- |
| Vehicle mass | Total vehicle weight | 24000 kg |
| Vehicle length | Total length of the vehicle | 11575 mm |
| Vehicle width | Total width of the vehicle | 2500 mm |
| Vehicle height | Total height of the vehicle | 3855 mm |
| Wheel base | Distance between front and rear axles | 3505 mm |
| Distance from CG to front wheel axle | Horizontal distance from CG to front axle | 1935 mm |
| Center of mass height | Vertical height of the center of mass | 1650 mm |
| Tire specification | Tire model and load capacity | 385/95R25 170F |

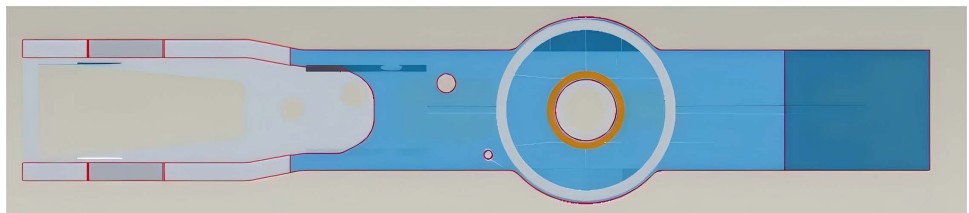

**Fig 2. Schematic diagram of the chassis structure.**

plate, incorporating weight-reduction holes in non-load-bearing areas, and reinforcing local load-bearing areas with additional plates.The main material used for the chassis structure is Q960E, with a safety factor $n = 1.33$. The allowable stress is calculated as $[\sigma] = \frac{0.5\sigma s + 0.35\sigma b}{n} = 640MPa$. A schematic diagram of the chassis structure is shown in Fig 2.

## 2.3. Suspension design

The suspension system encompasses all load-transferring connection mechanisms between the chassis and axle. While the elastic properties of hydro-pneumatic springs can mitigate road impacts, they cannot fully dissipate the impact energy. To address the dual requirements of high load-carrying capacity and driving comfort, a four-point independently controllable hydro-pneumatic suspension hydraulic system was developed. This system, combined with a 260 mm travel suspension, enhances the vehicle's adaptability and off-road capabilities. The optimization of comfort was achieved through improvements in both the stiffness and damping characteristics of the vehicle.

The stiffness of hydro-pneumatic suspension can be determined based on thermodynamic equations:

$$P_0 V_0^n = P_1 V_1^n \tag{1}$$

Among these: $P_0$ —— Mid-pressure

$V_0^n$ —— Mid-volume

$P_1$ —— Dynamic pressure

$V_1^n$ —— Dynamic volume

Following modifications to the cylinder dimensions and accumulator volume, the system necessitates recalculation. Fig 3 illustrates the displacement-pressure variation curve of the hydro-pneumatic suspension system.

In addition to calculating the suspension system parameters, factors influencing damping require adjustment to achieve the desired comfort level. During upward movement, the volume of the suspension cylinder's closed cavity decreases, causing excess

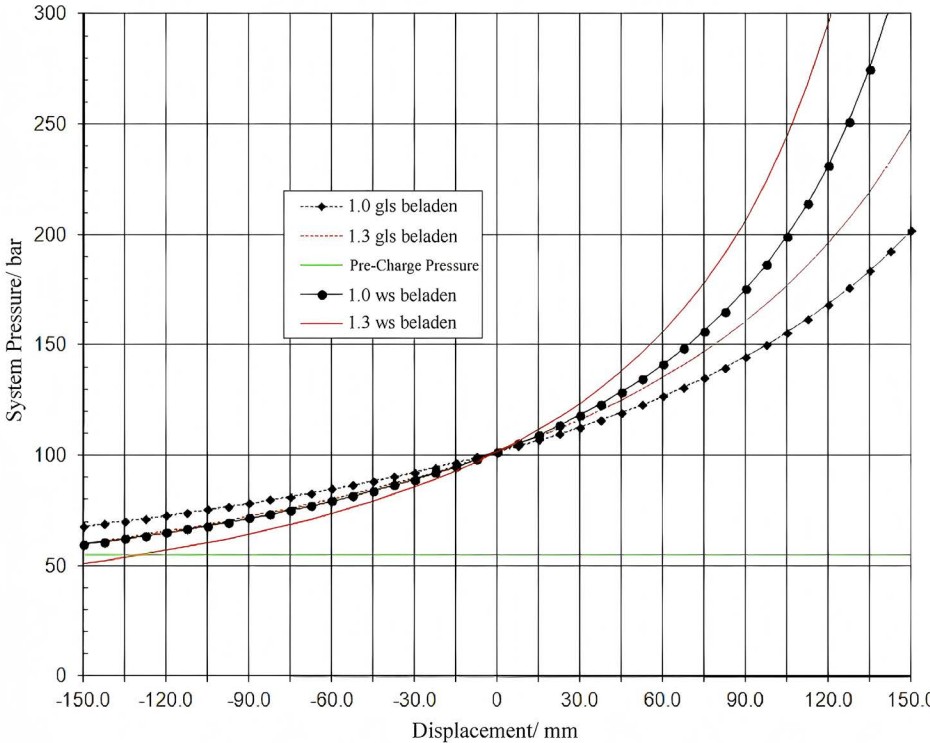

**Fig 3. Hydro-pneumatic suspension displacement-pressure variation curve.**

fluid to enter the accumulator, resulting in increased accumulator pressure. When the piston speed x>0, the piston moves upward, with only the damping hole permitting flow, generating damping force to rapidly reduce suspension vibration and dissipate kinetic energy. When the piston speed x<0, the piston moves downward through the one-way valve and damping hole, producing damping force. Due to the one-way valve's large diameter, it swiftly releases oil from the annular cavity inside the cylinder rod to the rod chamber, which connects to the accumulator. Consequently, oil rapidly enters the accumulator, leading to increased accumulator pressure, enabling it to absorb impact forces from the vehicle on rough terrain. Fig 4 illustrates the design structural diagram.

The suspension cylinder design incorporates a one-way valve to facilitate rapid oil flow release. During downward movement, the system minimizes damping force, while during upward movement, it increases damping force to swiftly reduce vibration and enhance passenger comfort. The damping holes are strategically positioned to ensure that under specific extreme conditions, when the cylinder reaches its upward limit, the damping mechanism fully closes, providing additional protection to the cylinder.

## 3. Optimization design of the steering control system

### 3.1. Analysis of the vehicle steering system

To enhance the vehicle's maneuverability and flexibility, a two-axle multi-mode electro-hydraulic proportional steering system was developed. The front axle employs mechanical steering, while the rear axle utilizes precise electro-hydraulic proportional control steering technology. This configuration enables an optimal turning radius, multi-mode steering, and improved vehicle maneuverability in confined spaces. A coordinate system is established with the center point of the front axle as the origin. The X-axis points in the forward direction, parallel to the ground, while the Y-axis points in the direction of the driver, parallel to the front axle. The Z-axis direction is determined using the right-hand rule, as illustrated in Figs 5 and 6.

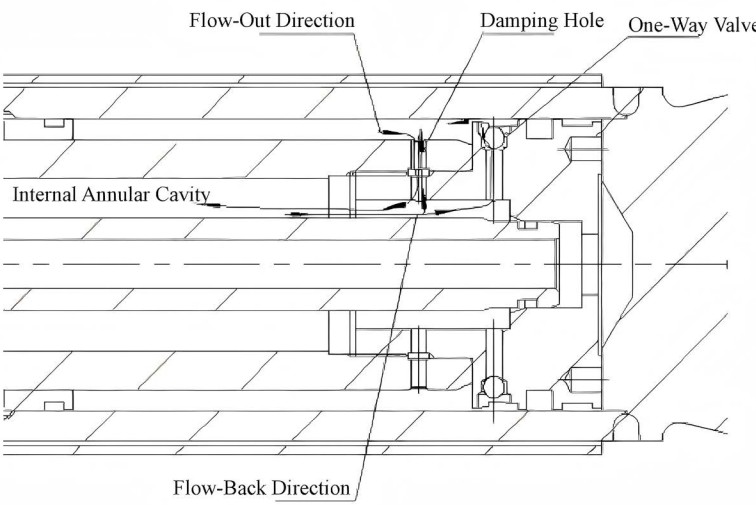

**Fig 4. Suspension Cylinder Damping Structure Diagram.**

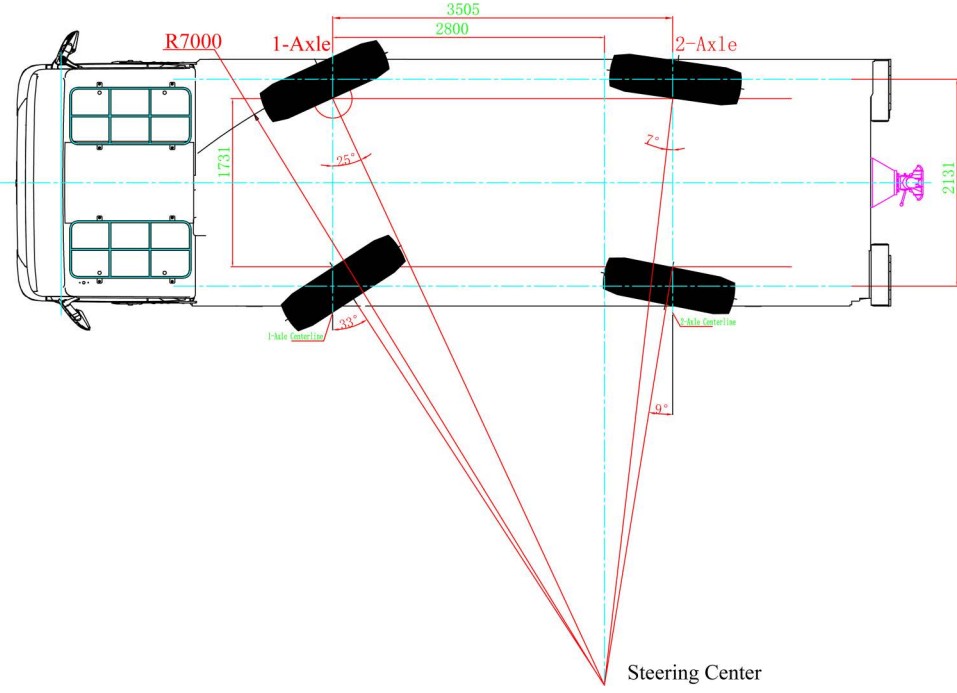

**Fig 5. Normal driving mode.**

During standard vehicle operation, the kingpin inclination angle $\beta = 6°$, and the kingpin center distance $M = 1598mm$. The relevant calculation formula is as follows:

$$R_{\min} = \frac{L}{\sin \theta_{0\,\max}} + a \qquad (2)$$

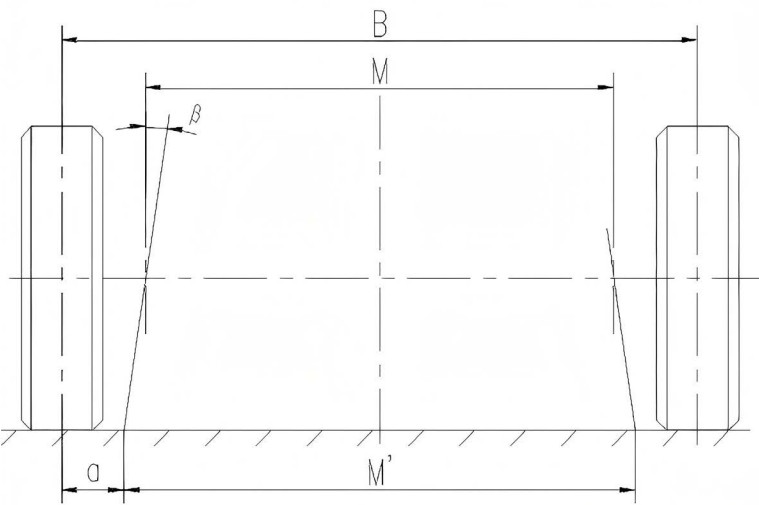

**Fig 6. Tire steering angle diagram.**

$$M' = M + 2 \times R_d \times \tan \beta \tag{3}$$

$$\cot \theta_{0\,max\,1} - \cot \theta_{i\,max\,1} = \frac{M'}{L} \tag{4}$$

Among these: $M'$ —— The distance between the ground intersection points of the two kingpin centerlines
$a$ —— Kingpin offset distance
$\theta_{0\,max\,1}$ —— Maximum Steering Angle of the Outer Wheel on the 1-Axle
$\theta_{i\,max\,1}$ —— Maximum Steering Angle of the Inner Wheel on the 1-Axle
Based on the aforementioned formula, the calculations yield:

$$\theta_{0\,max\,1} = \sin^{-1} \frac{L}{R_{min\,1} - a} = \sin^{-1} \frac{2800}{7000 - 197} = 24.3° \tag{5}$$

$$M_1' = 1598 + 2 \times 662 \times \tan 6° = 1737mm \tag{6}$$

$$a = \frac{B - M'}{2} = \frac{2131 - 1737}{2} = 197mm \tag{7}$$

$$\theta_{i\,max\,1} = \cot^{-1} \left( \cot \theta_{0\,max\,1} - \frac{M'}{L} \right) = \cot^{-1} \left( \cot 24.3° - \frac{1737}{2800} \right) = 32.1° \tag{8}$$

Regarding the overall configuration, the maximum outer wheel steering angle of Axis 1 is established at $\theta_{0\,max\,1} = 25°$ , while the maximum inner wheel steering angle of Axis 1 is set to $\theta_{i\,max\,1} = 32.5°$. Under these specified parameters, the calculated theoretical turning radius measures 6936 mm.

In the scenario where the rear wheels steer independently, the theoretical steering center should be positioned along the centerline between Axis 1 and Axis 2 to minimize the turning radius. Due to constraints imposed by the vehicle's overall width, the maximum steering angle of the outer wheel on Axis 2 is established at $\theta_{0\max2} = 24°$. Under these parameters, the distance from the instantaneous steering centerline to the minimum distance from Axis 1 is calculated to be 1800 mm. Consequently, the steering center distance from Axis 1 is determined to be $L_{1m} = 1800mm$, with the maximum steering angle of the outer wheel on Axis 2 set at $\theta_{0\max2} = 24°$, and the maximum steering angle of the inner wheel on Axis 2 set at $\theta_{i\max2} = 32°$. Given these specifications, the minimum turning radius achieved is 4850 mm, as illustrated in Fig 7.

### 3.2. Linear 2DOF model

The evaluation of vehicle stability primarily relies on the yaw angular velocity. To control the vehicle's stability, the control system generates the corresponding yaw moment through power distribution to each wheel. Based on this principle, a linear two-degree-of-freedom (2 DOF) model is established, as illustrated in Fig 8.

The differential equations of motion for the linear two-degree-of-freedom (2 DOF) model are expressed as follows:

$$(k_1 + k_2)\beta + \frac{1}{\mu}(ak_1 + bk_2)\omega_r - k_1\delta = m(\upsilon\mu\omega_r)$$
$$(ak_1 + bk_2)\beta + \frac{1}{\mu}(a^2k_1 + b^2k_2)\omega_r - ak_1\delta = I_z\omega_r$$

(9)

Among these: $m$ —— The mass of the vehicle

$I_z$ —— The moment of inertia of the vehicle around the Z-axis

$\omega_r$ —— The vehicle's yaw angular velocity

$k_1, k_2$ —— The stiffness of the front and rear wheels

$a, b$ —— The distance from the vehicle's center of gravity to the front axle and rear axle

$\mu, \upsilon$ —— The vehicle's velocity in the X-axis direction and the Y-axis direction

$\beta$ —— The lateral displacement angle of the center of gravity

$\delta$ —— The steering angle of the front wheels

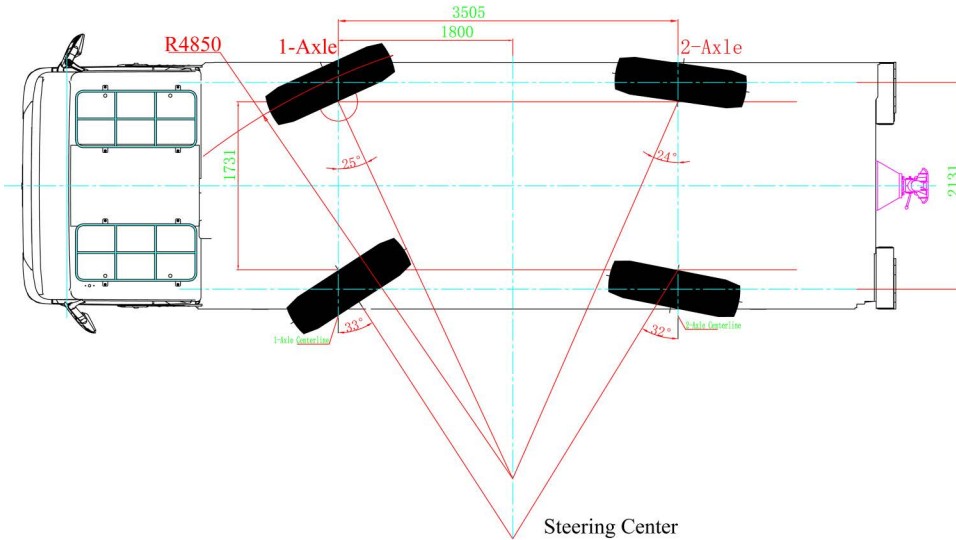

**Fig 7. Rear Independent Steering Mode.**

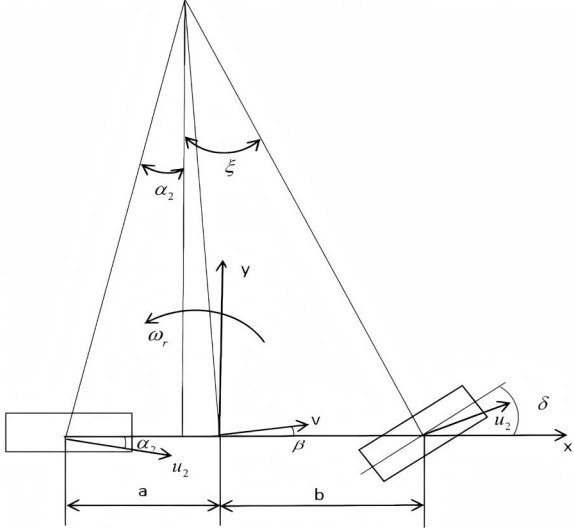

**Fig 8. Linear 2 DOF Model.**

In the linear two-degree-of-freedom (2 DOF) model, when $\mu = 0$ and $\omega_r = 0$, the vehicle achieves a steady-state condition. The equation for determining the ideal vehicle yaw rate is expressed as:

$$\omega_r = \frac{\mu}{L\left(1 + K\mu^2\right)}\delta$$

(10)

Among these: $K$ —— Stability coefficient, $K = \frac{m}{(a+b)^2} \times \left(\frac{a}{k_r} - \frac{b}{k_f}\right)$

$L$ —— Axle distance

To express equations (9) and (10) in the form of state-space equations, we identify the state variables $X = \begin{bmatrix} \beta & \omega_r \end{bmatrix}^T$, input variables $U = \begin{bmatrix} \delta_f & \delta_R \end{bmatrix}^T$, and output variables $Y = \begin{bmatrix} \beta & \omega_r \end{bmatrix}^T$. Consequently, we derive the following:

$$X = AX + BU$$
$$Y = CX$$

(11)

Among these:

$$A = \begin{bmatrix} \frac{k_f + k_r}{m\mu} & \frac{ak_f - bk_r}{mV^2} - 1 \\ \frac{ak_f - bk_r}{I_z} & \frac{a^2 k_f \mp k_r}{I_z\mu} \end{bmatrix} \quad B = \begin{bmatrix} \frac{-k_f}{m\mu} & \frac{-k_r}{m\mu} \\ \frac{ak_f}{I_z} & \frac{bk_r}{I_z} \end{bmatrix} \quad C = \begin{bmatrix} \frac{-k_f}{m\mu} & \frac{ak_f}{I_z} \end{bmatrix}^{-1}$$

### 3.3. PID optimized control model

This control model employs the lateral offset of the center of gravity as the primary control variable. A PID control system adjusts the longitudinal forces of individual wheels to equilibrate the yaw moment required during steering. Fig 9 depicts the control model schematically.

The control model of the XZJ5240JQZ30 all-terrain rescue lift vehicle utilizes the steering wheel angle and speed as initial input conditions, establishing a linear 2DOF model to calculate the ideal vehicle slip angle while providing real-time feedback on the linear relationship between the vehicle slip angle and active lateral acceleration. The vehicle's operating

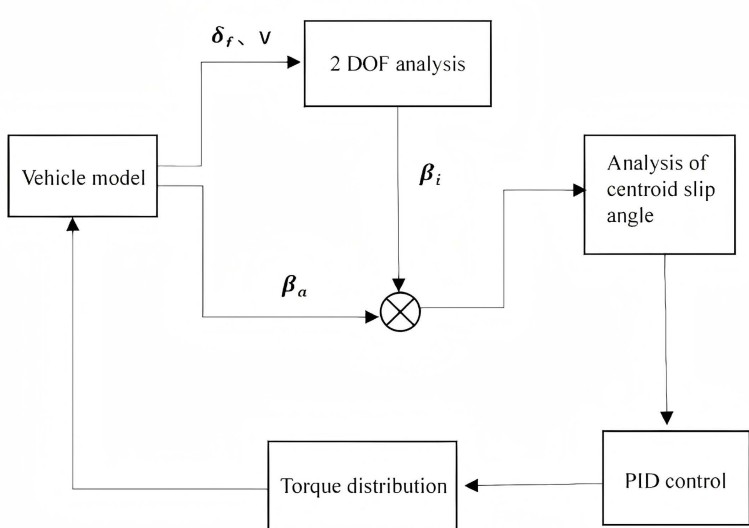

**Fig 9. Vehicle PID control model.**

state is determined by the deviation between the actual and ideal vehicle slip angle, thereby activating the PID control system. When the vehicle exhibits oversteering, the deviation in vehicle slip angle triggers the control system, which adjusts the vehicle's motion by generating the required additional yaw moment and distributing it among the wheels. This torque distribution enhances the vehicle's stability during maneuvering.

To determine suitable PID gain parameters for the XZJ5240JQZ30 all-terrain rescue lift vehicle, the trial-and-error method was employed for parameter tuning. Initially, the integral and derivative constants were set to zero, and the proportional coefficient was gradually increased until oscillations appeared in the response curve. Adjustments continued until the oscillations disappeared, allowing the proportional constant to be determined. Following this approach, the integral and derivative constants were then tuned sequentially. Ultimately, the optimized PID parameters were obtained as P = 8, I = 5, and D = 1.

### 3.4. Steering control method for all-terrain rescue lift vehicle

During high-speed steering maneuvers of an all-terrain rescue lift vehicle, a negative yaw moment is necessary to counteract the lateral acceleration generated by the steering input and maintain vehicle stability. In this scenario, increasing the power to the outer wheels while reducing power to the inner wheels proves most effective. Consequently, the steady-state steering condition of the all-terrain rescue lift vehicle is determined based on the front wheel steering angle and the yaw rate deviation, with four-wheel drive power allocation adjusted accordingly. The formula for the effective deviation is expressed as follows:

$$e = \Delta\delta_r - \Delta\delta_r^+ \quad \Delta\delta_r > \Delta\delta_r^+$$
$$e = 0 \quad \Delta\delta_r^- \leq \Delta\delta_r \leq \Delta\delta_r^+$$
$$e = \Delta\delta_r - \Delta\delta_r^- \quad \Delta\delta_r < \Delta\delta_r^-$$

$$(12)$$

When $\Delta\delta_r^- \leq \Delta\delta_r \leq \Delta\delta_r^+$ and $e = 0$, the control system remains inactive. However, when $\Delta\delta_r > \Delta\delta_r^+$ and $\Delta\delta_r < \Delta\delta_r^-$, the control system activates to enhance driving stability through power distribution. Fig 10 illustrates the integrated simulation control model for the steering control system.

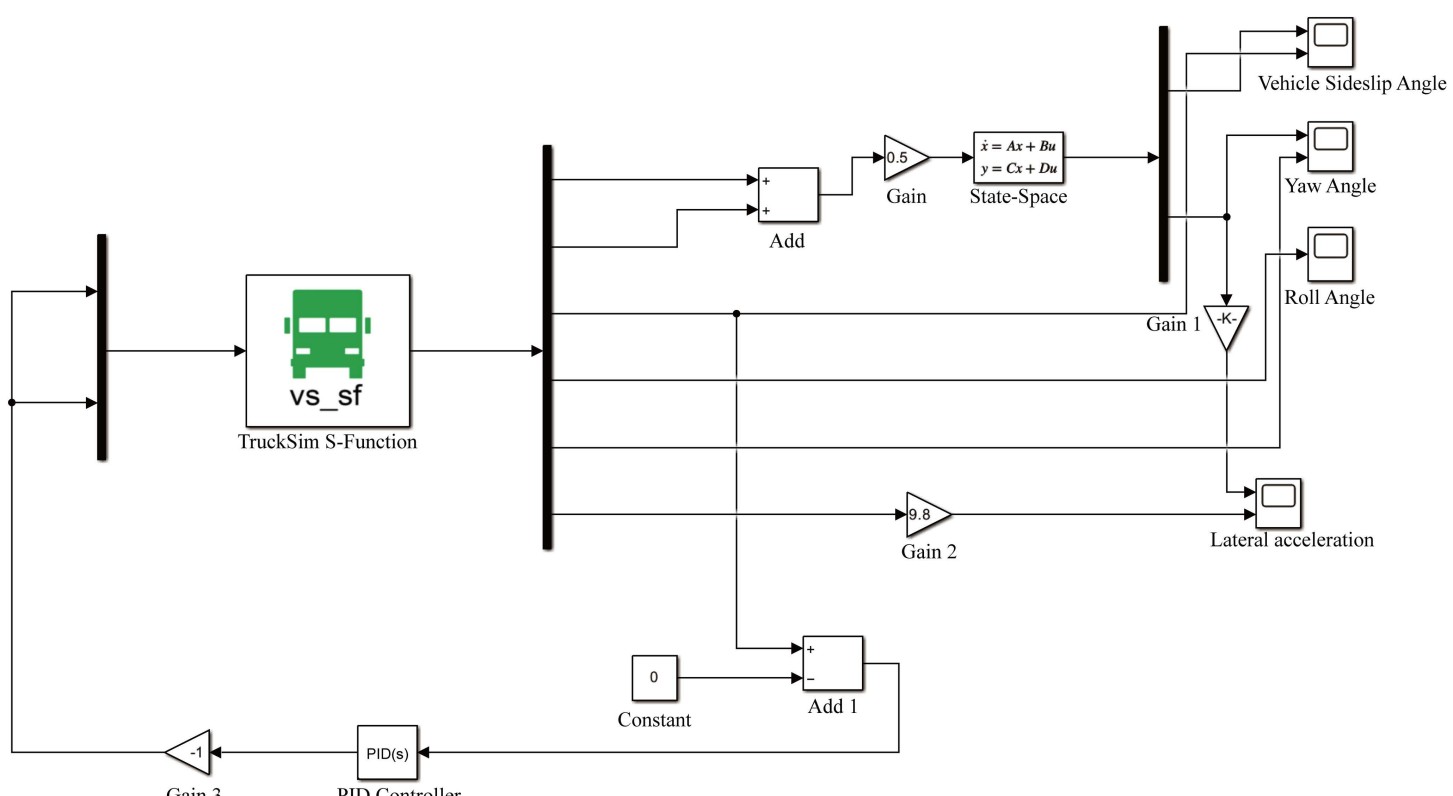

## 4. Numerical model

The open-loop input simulation study of the driver is conducted by determining the steering angle of the front wheels. To more accurately simulate the actual driving conditions and operational environment of the all-terrain rescue lift vehicle, a joint simulation using TruckSim and Simulink is employed under the classic Double lane-change scenario. The study focuses on comparing active steering control at high-speed (70 km/h) and low-speed (30 km/h) conditions, analyzing the variations in the yaw rate and the lateral offset of the CG. Subsequently, the impact of the center of gravity height on low-speed four-wheel steering (4WS) under Double lane-change is analyzed, examining the influence of the designed system on vehicle stability.

### 4.1. Simulation parameter settings

For the simulation process, a 2A Euro Cab-over type model vehicle, which closely resembles the overall appearance of the all-terrain rescue hoisting vehicle, was selected for numerical simulation. Based on the actual data of the XZJ5240JQZ30 all-terrain rescue lift vehicle, various fundamental parameters of the model vehicle were adjusted in Trucksim. The XZJ5240JQZ30 all-terrain rescue lift vehicle is equipped with an MC07H.33–60 engine and a 12AS1215S0 automated manual transmission. According to its actual parameters, the vehicle's control air intake in Trucksim was set to ≥ 1250 kg/h, with a pressure difference of ≤ 45 kPa, a maximum input torque of 1200 Nm, and a maximum input speed of 2650 rpm. The remaining specific vehicle parameters are shown in Table 2.

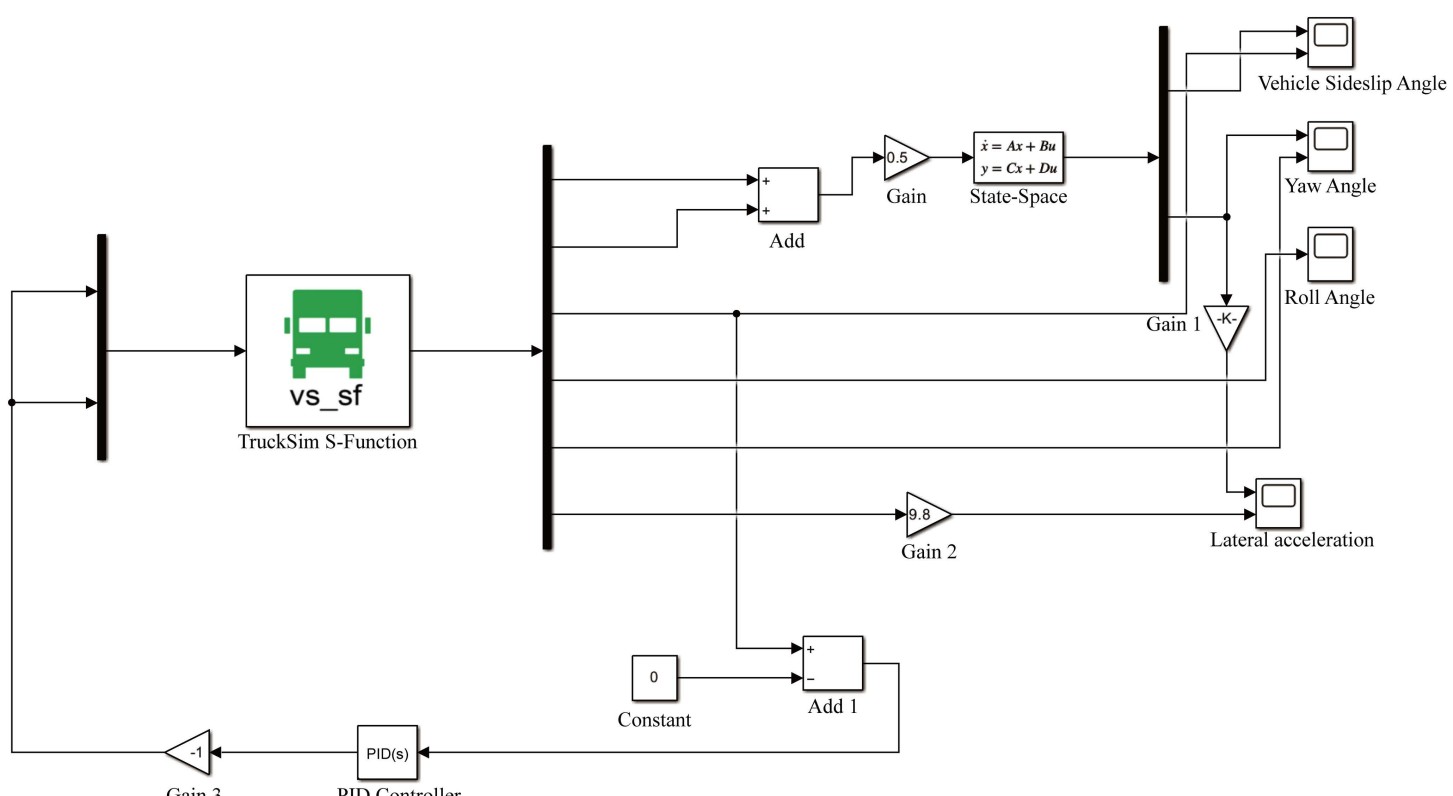

**Fig 10. Integrated simulation control structure diagram for the steering control system.**

To more accurately reflect the motion state of the XZJ5240JQZ30 all-terrain rescue lift vehicle on mine rescue roads, the simulation conditions were set to a Double lane-change scenario based on actual mine rescue road conditions. The road surface material was defined as dry asphalt, and the road adhesion coefficient was set to µ = 0.85.

## 4.2. Validation of simulation

To verify the compatibility between the model built in Simulink based on theoretical formulas and the built-in model in Trucksim, it was necessary to perform a simulation validation of the ideal 2DOF model before conducting the Trucksim and Simulink co-simulation experiments. This ensures the reliability of the established XZJ5240JQZ30 all-terrain rescue lift vehicle model. Under the classic Double lane-change scenario, a sinusoidal input was applied for simulation validation, and the variations in the centroid sideslip angle and yaw rate were monitored. The results, shown in Fig 11, indicate that the motion trajectory of the linear 2DOF model built in Simulink closely follows the driving trajectory of the Trucksim model, effectively verifying the feasibility of the vehicle model.

## 4.3. Analysis of vehicle stability at high speeds

In emergency situations, the rescue team needs to drive the all-terrain rescue lift vehicle at high speeds while ensuring both safety and maneuverability. Therefore, in this simulation experiment, the vehicle speed was set to 70 km/h to

**Table 2. Vehicle model parameters.**

| Simulation parameters | Parameters setting |
| --- | --- |
| Vehicle quality | 24000 kg |
| Vehicle length | 11575 mm |
| Vehicle width | 2500 mm |
| Tyre height | 635 mm |
| Steering damping | 4.0 Nm·s/deg |
| Hys reference angle | 0.5 deg |
| Spin inertia for each side | 10.0 kg·m$^2$ |

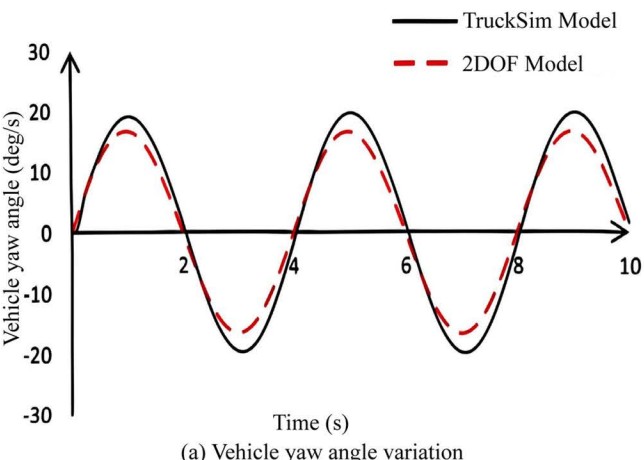

(a) Vehicle yaw angle variation

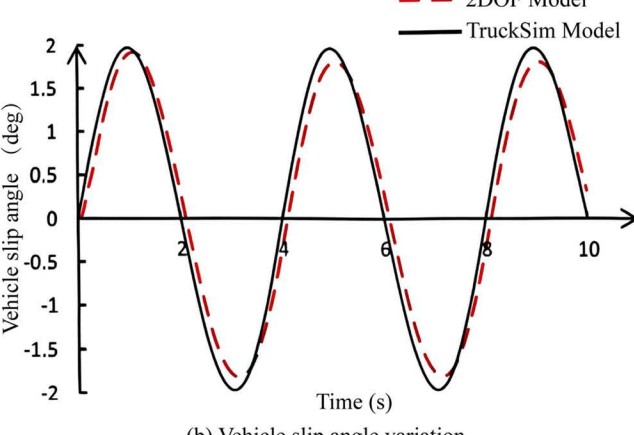

(b) Vehicle slip angle variation

**Fig 11. Simulation verification of the vehicle model.**

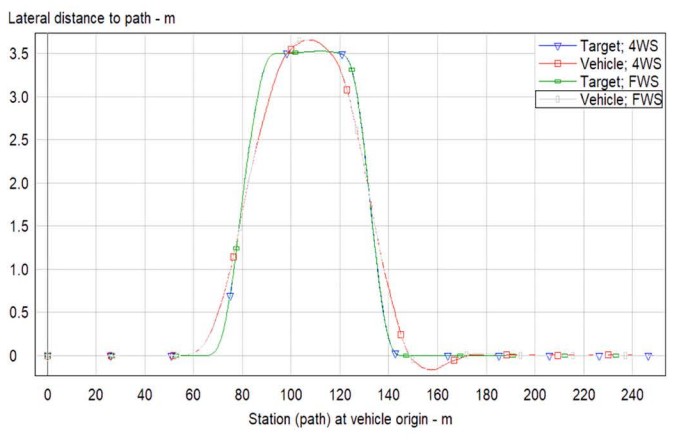

compare and analyze the performance parameters of the four-wheel steering (4WS) all-terrain rescue lift vehicle with PID control and the FWS all-terrain rescue lift vehicle without PID control. The results are shown in Fig 12.

As illustrated in Fig 12a, during high-speed driving, the trajectory of the 4WS vehicle with PID control exhibits a slight deviation compared to the FWS vehicle without PID control from the initial steering phase until 3.2 seconds. However, from 3.2 seconds until the conclusion of the simulation, the 4WS vehicle with PID control closely adheres to the intended path. In high-speed driving scenarios, PID control demonstrates a substantial stabilizing effect on the all-terrain rescue lift vehicle, particularly on curved road sections. To conduct a quantitative analysis of the simulation results, the peak values of the steering wheel angle, vehicle slip angle, and vehicle yaw angle were extracted and are presented in Table 3.

A comprehensive analysis of Fig 12b–d, and Table 3 demonstrates that under high-speed driving conditions, the 4WS vehicle effectively controls the vehicle slip angle, significantly reducing its positive and negative peak values compared to the FWS vehicle. Steering perception, being a subjective experience for the driver during vehicle turning, can be reflected in the vehicle's yaw angle response during cornering. In the given Double lane-change scenario, the yaw angle of the 4WS vehicle closely matches that of the FWS vehicle, ensuring consistent road feel for the driver, regardless of PID control application. However, at high speeds, the 4WS vehicle's steering wheel angle response does not show substantial

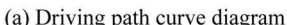

(a) Driving path curve diagram

(b) Steering wheel angle

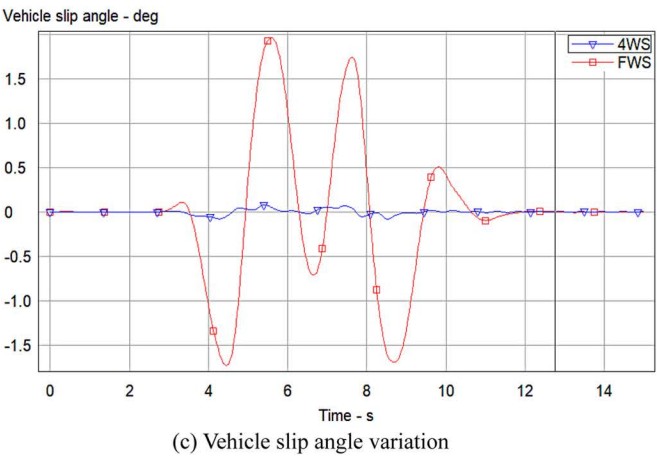

(c) Vehicle slip angle variation

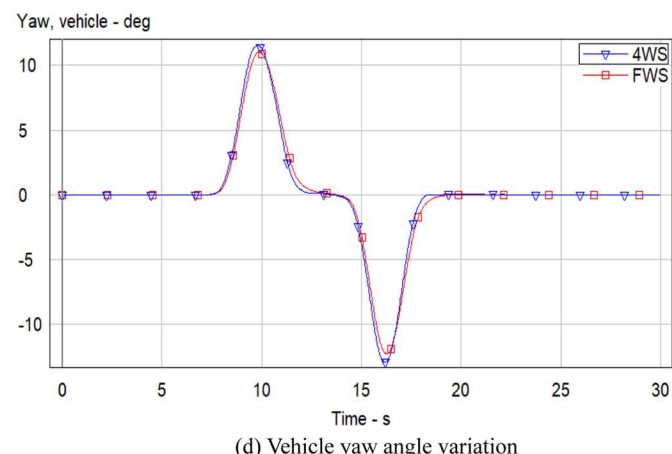

(d) Vehicle yaw angle variation

**Fig 12. Parameter curve changes at high speed.**

Table 3. Peak Data at High Speed.

| Variable Name | 4WS | FWS |
|---|---|---|
| Steering wheel angle (deg) | 39, -41 | 38, -41 |
| Vehicle slip angle (deg) | 0.2, -0.2 | 1.9, -1.7 |
| Vehicle yaw angle (deg/s) | 12, -14 | 11, -13 |

improvements over the FWS vehicle, indicating that the effect of independent rear-wheel steering is not particularly pronounced under these conditions. This is primarily attributed to the vehicle's large mass, which results in a delayed response in independent rear-wheel steering control during high-speed driving.

### 4.4. Analysis of vehicle stability at low speeds

Under typical operational conditions, the all-terrain rescue lift vehicle primarily functions at reduced velocities. Consequently, this simulation experiment set the vehicle speed to 30 km/h to compare and analyze the performance parameters of the four-wheel steering (4WS) all-terrain rescue lift vehicle with PID control against the FWS all-terrain rescue lift vehicle without PID control. The comparative results are presented in Fig 13.

As illustrated in Fig 13a, under low-speed driving conditions, the trajectory of the 4WS vehicle with PID control demonstrates superior alignment with the simulated path, exhibiting an enhanced tracking effect compared to high-speed scenarios. To quantitatively assess the simulation outcomes, the maximum values of the steering wheel angle, vehicle slip angle, and vehicle yaw angle were extracted and are presented in Table 4.

A thorough examination of Fig 13b–d, and Table 4 indicates that at lower velocities, the enhancement in vehicle slip angle is more pronounced, exhibiting notable decreases at multiple peak points. Consistent with previous findings, the yaw angle of both 4WS and FWS vehicles remains largely unchanged, ensuring a uniform road feel for the driver, irrespective of PID control application. Furthermore, the study observed a substantial reduction in the peak positive and negative values of the steering wheel angle for the 4WS vehicle with PID control. This observation suggests that during active steering, the 4WS all-terrain rescue lift vehicle can execute more efficient steering maneuvers without necessitating large steering wheel angles.

### 4.5. Analysis of the influence of center of mass height on vehicle stability

The all-terrain rescue lift vehicle examined in this study incorporates a chassis suspension with a maximum travel of 262 mm. While the long-travel suspension substantially improves the vehicle's off-road performance, it also impacts driving stability. Consequently, this simulation experiment was conducted at a velocity of 30 km/h utilizing the PID-controlled 4WS vehicle model. Under the Double lane-change scenario, two center-of-mass heights, 1650 mm and 1910 mm, were evaluated, with the results presented in Fig 14.

For a quantitative analysis of the simulation results, the maximum values of the steering wheel angle and vehicle slip angle were documented, as presented in Table 5.

A thorough examination of Fig 14 and Table 5 reveals notable distinctions in steering wheel angles between the two center-of-mass heights. The positive and negative peak values for the lower center-of-mass 4WS vehicle are considerably reduced compared to those of the higher center-of-mass vehicle. Furthermore, after 18 seconds of operation, the steering wheel angle of the higher center-of-mass vehicle demonstrates approximately 7 seconds of oscillation, with a maximum difference of 15 degrees. The overall variation trend of the vehicle slip angle remains consistent for both high and low center-of-mass vehicles. However, the slip angle of the lower center-of-mass vehicle exhibits greater stability and improved convergence relative to the higher center-of-mass vehicle. These observations suggest that a lower center of mass enhances vehicle steering stability and reduces the steering wheel angle required for effective maneuvering.

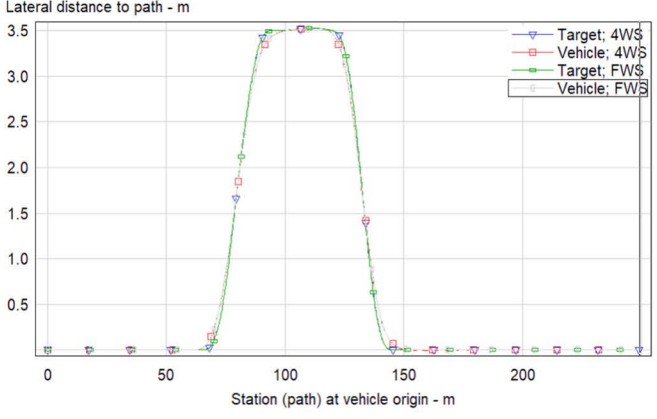

(a) Driving path curve diagram

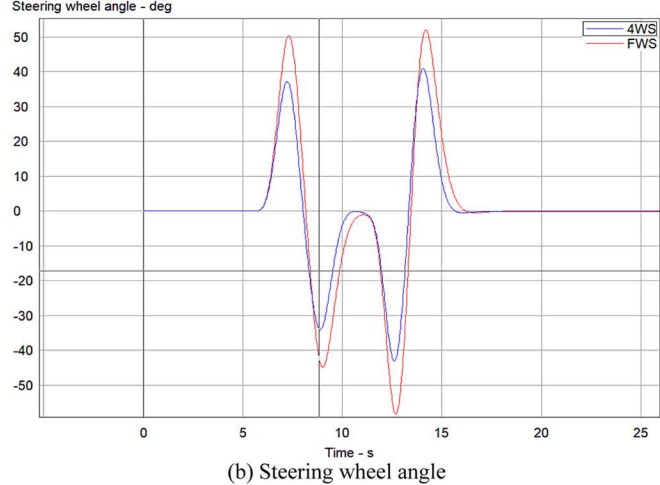

(b) Steering wheel angle

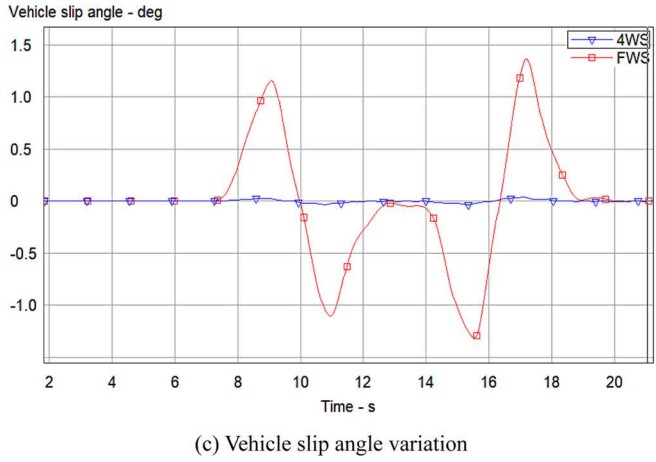

(c) Vehicle slip angle variation

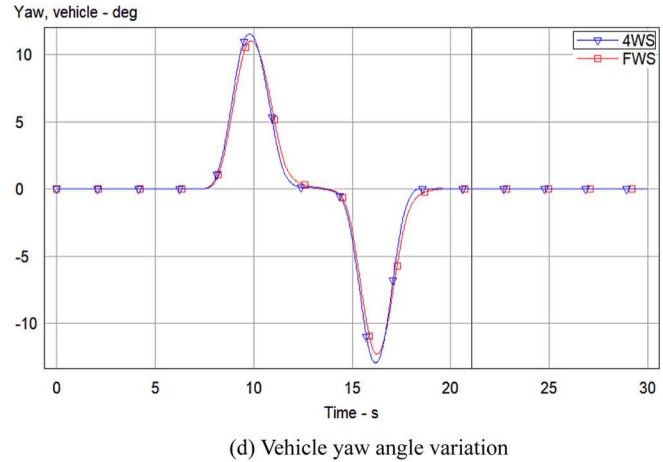

(d) Vehicle yaw angle variation

**Fig 13. Parameter curve changes at low speed.**

**Table 4. Peak data at low speed.**

| Variable Name | 4WS | FWS |
|---|---|---|
| Steering wheel angle (deg) | 42, -44 | 53, -60 |
| Vehicle slip angle (deg) | 0.1, -0.1 | 1.4, -1.3 |
| Vehicle yaw angle (deg/s) | 12, -14 | 11, -13 |

## 5. Steering control experiment of the all-terrain rescue lift vehicle

Following the optimization of the vehicle structure and steering system, as well as the analysis of numerical simulation results, a steering control experiment is conducted to further evaluate the practical application value of the PID-optimized control system in enhancing vehicle stability. The experiment is carried out on emergency rescue roads in the auxiliary shaft area of the Meihua Mine using the XZJ5240JQZ30 all-terrain rescue lift vehicle.

The experiment recorded the theoretical steering angle, actual vehicle steering angle, and steering angle error, with the results depicted in Fig 15. The experimental findings demonstrate that the improved vehicle's actual steering angle

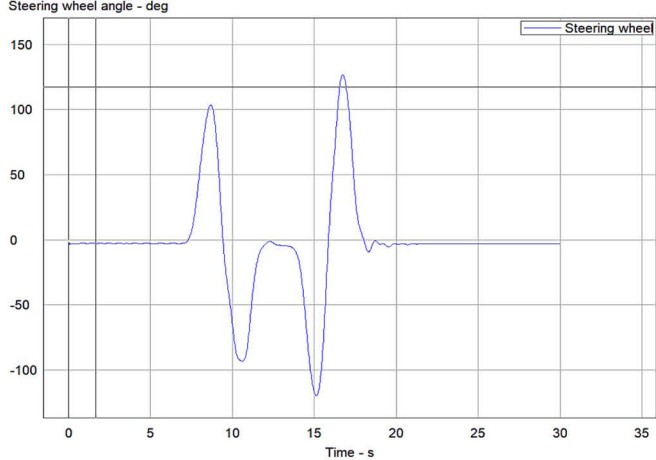

(a) Steering wheel angle of lower center-of-mass
(b) Steering wheel angle of higher center-of-mass

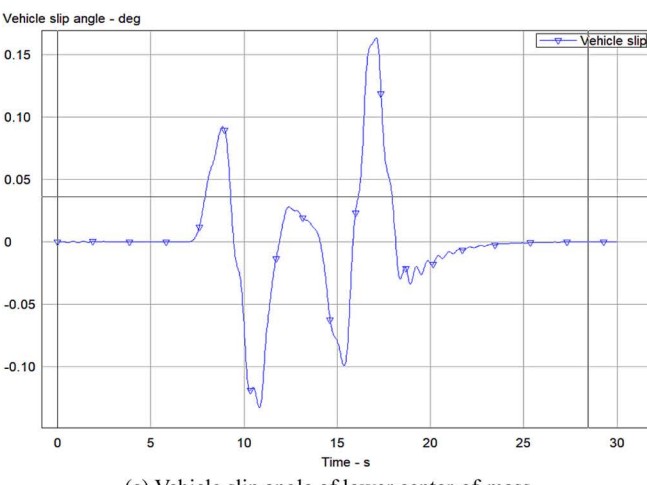
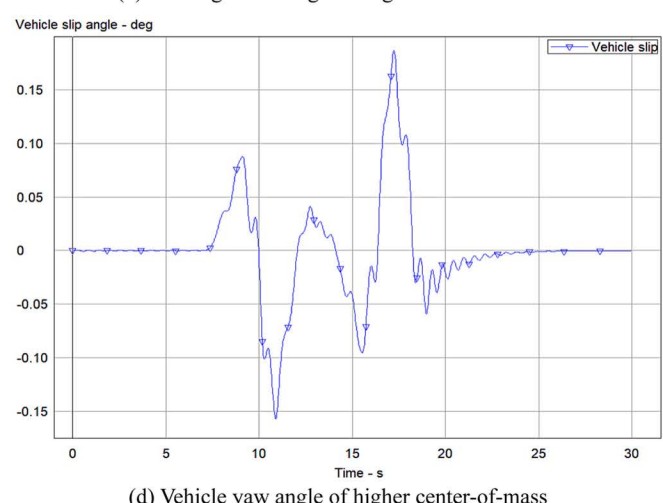

(c) Vehicle slip angle of lower center-of-mass
(d) Vehicle yaw angle of higher center-of-mass

**Fig 14. Parameter curve changes at different center-of-mass heights.**

**Table 5. Peak data at different center-of-mass heights.**

| Variable Name | Higher center-of-mass | Lower center-of-mass |
|---|---|---|
| Steering wheel angle (deg) | 145.5, -127.2 | 126.8, -119.8 |
| Vehicle slip angle (deg) | 0.18, -0.16 | 0.17, -0.13 |

generally satisfies practical requirements, exhibiting minimal steering error and response delay. The actual steering angle closely approximates the theoretical values, with the steering angle error reaching a negative peak of -3.5% at 20 seconds and a maximum positive peak of 2.5% at 245 seconds. These results corroborate the effectiveness of the PID-optimized control model proposed in this study in enhancing the stability of the all-terrain rescue lift vehicle.

Concurrently, the optimized chassis and suspension structure proposed in this study underwent testing during the experiment. Through comparative analysis, the road adaptability and terrain traversal capability of the XZJ5240JQZ30 all-terrain rescue lift vehicle were assessed. Post-optimization, the vehicle's minimum turning diameter was reduced to 9.7 m, with the maximum outer wheel steering angle of the second axle reaching 24 deg and the maximum inner wheel steering angle reaching 32 deg.

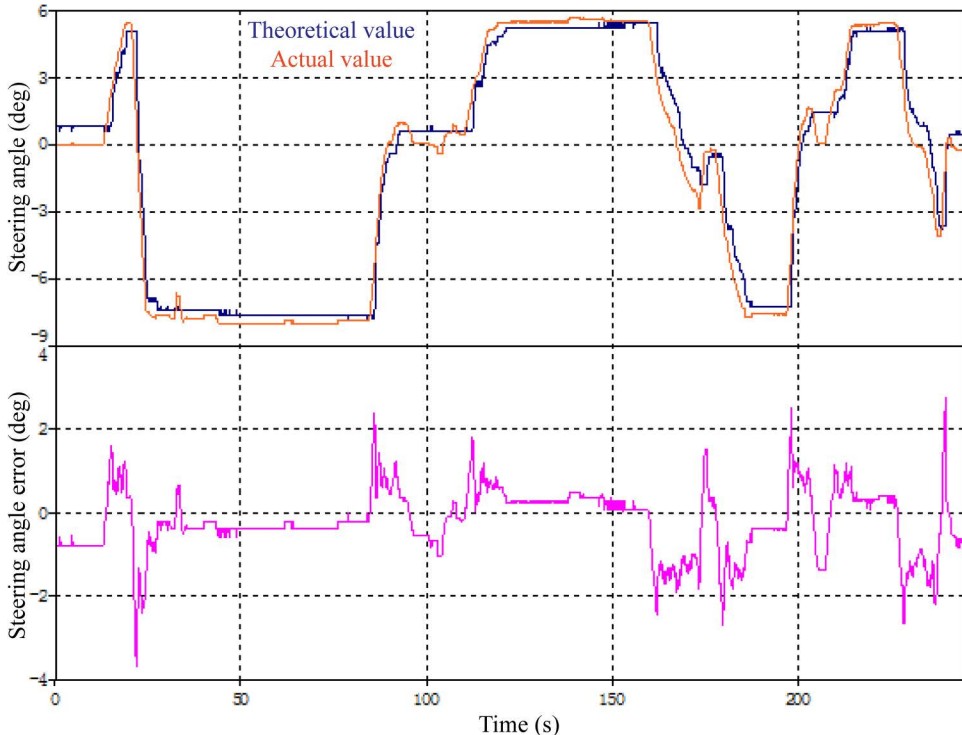

**Fig 15. Parameter curve changes in field tests.**

## 6. Results and discussion

This study optimizes the vehicle structure and steering system of the XZJ5240JQZ30 all-terrain rescue lift vehicle. A PID-optimized 4WS control system is implemented, which adjusts the longitudinal forces of individual wheels to balance the required yaw moment during steering, thereby enhancing both steering stability and off-road capability. Numerical simulations were conducted using TruckSim and Simulink to analyze the vehicle's performance under various conditions in the Double lane-change scenario. Additionally, field experiments are carried out to test and validate the feasibility of the proposed control system in real-world rescue operations.

The findings demonstrate that the PID-optimized control system substantially enhances vehicle stability. In high-speed scenarios, the PID-controlled 4WS system effectively minimizes the vehicle slip angle. In comparison to the conventional FWS system, the proposed control model maintains a more consistent vehicle trajectory, ensuring improved maneuverability and safety during rescue operations. In low-speed conditions, the 4WS system not only suppresses slip angle fluctuations but also decreases the required steering wheel angle, thus enhancing vehicle control in narrow and complex terrains. Additionally, the study examined the influence of center-of-mass height on stability, revealing that a lower center of mass improves steering stability and reduces steering wheel angle variations during turns. Based on these observations, steering control experiments were conducted on the all-terrain rescue lift vehicle, further corroborating the efficacy of the proposed PID control model. Experimental data revealed that the actual steering angles closely approximated theoretical values, with minimal error and response delay. This validates the viability of the PID-optimized 4WS system in practical applications, offering both theoretical and empirical references for enhancing all-terrain rescue lift vehicles.

Despite these promising results, the study presents certain limitations. The research on the control system remains insufficient, as it does not explicitly demonstrate the system's approach to addressing nonlinear dynamics in vehicle motion. Furthermore, the real-time implementation of the control algorithm on an onboard system has not been thoroughly

investigated. To address these constraints, future research will concentrate on advanced control techniques for nonlinear systems and the real-time implementation of control algorithms on onboard systems. This focus aims to ensure practical applicability in emergency rescue scenarios.

## 7. Main conclusions

Through the optimization of the chassis and suspension structure of the XZJ5240JQZ30 all-terrain rescue lift vehicle, a long-travel hydro-pneumatic suspension system was developed. This enhancement enables the vehicle to adapt more effectively to the challenging road conditions in mining areas, thereby substantially improving its off-road performance.

The PID-optimized four-wheel steering (4WS) control system proposed in this study significantly enhances the steering stability of the XZJ5240JQZ30 all-terrain rescue lift vehicle. This improvement is demonstrated through a reduction in vehicle slip angle and enhanced trajectory tracking capabilities under both high-speed and low-speed conditions. Furthermore, the research indicates that a lower center of mass contributes to improved vehicle maneuverability and stability. These findings offer valuable insights for future optimization of vehicle structure and steering control systems in similar applications.

Real-world application tests demonstrated that the PID-optimized 4WS control system exhibits excellent performance in practical scenarios. The observed steering angles closely aligned with the theoretical values, showing minimal error and response delay. These results validate the system's feasibility for implementation in actual rescue operations.

Subsequent research endeavors will concentrate on advancing control methodologies to more effectively manage the nonlinear dynamics inherent in rescue vehicles. Moreover, investigations will be undertaken to explore the real-time implementation of the control system on embedded in-vehicle platforms. Further evaluations under diverse environmental conditions will be conducted to validate the efficacy of the proposed system in authentic rescue scenarios.

### 7.1. Nomenclature

| | |
|---|---|
| $M^1$ | The distance between the intersection points of the two kingpin centerlines and the ground [mm] |
| $I_Z$ | Moment of inertia [kg/m³] |
| $\omega_r$ | Yaw rate [deg/s] |
| $k_f$ | Front wheel stiffness [N/m] |
| $k_r$ | Rear wheel stiffness [N/m] |
| $\beta$ | Center of mass slip angle [deg] |
| $\delta$ | Front wheel steering angle [deg] |
| $\mu$ | Vehicle velocity in the x-axis direction [km/h] |
| $\upsilon$ | Vehicle velocity in the y-axis direction [km/h] |

## Author contributions

**Conceptualization:** Dongmei Tian.

**Data curation:** Jiayun Wang.

**Funding acquisition:** Dongmei Tian.

**Investigation:** Jimao Shi.

**Project administration:** Weiyu Qu.

**Software:** Baiyou Xu.

**Validation:** Shouyi Wang.

**Writing – original draft:** Jian Yao.

**Writing – review & editing:** Weiyu Qu.

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
