## [Decision Letter · Decision Letter 0]

6 Jan 2025

PONE-D-24-59867Study on optimization control of rescue hoist vehicle safety performance based on state feedback: A research on the XZJ5240JQZ30 All-Terrain Rescue Lift VehiclePLOS ONE

Dear Dr. Tian,

Thank you for submitting your manuscript to PLOS ONE. After careful consideration, we feel that it has merit but does not fully meet PLOS ONE’s publication criteria as it currently stands. Therefore, we invite you to submit a revised version of the manuscript that addresses the points raised during the review process.

Please submit your revised manuscript by Feb 20 2025 11:59PM. If you will need more time than this to complete your revisions, please reply to this message or contact the journal office at plosone@plos.org . Please include the following items when submitting your revised manuscript:

We look forward to receiving your revised manuscript.

Kind regards,

Lei Zhang, PhD

Academic Editor

PLOS ONE

Journal Requirements:

Additional Editor Comments:

This study is generally well prepared, but there are still some places that need further revisions or clarifications.

Reviewers' comments:

Reviewer's Responses to Questions

**Comments to the Author**

1. Is the manuscript technically sound, and do the data support the conclusions?

Reviewer #1: Yes

Reviewer #2: Partly

2. Has the statistical analysis been performed appropriately and rigorously? 

Reviewer #1: No

Reviewer #2: N/A

3. Have the authors made all data underlying the findings in their manuscript fully available?

Reviewer #1: No

Reviewer #2: Yes

4. Is the manuscript presented in an intelligible fashion and written in standard English?

Reviewer #1: Yes

Reviewer #2: No

5. Review Comments to the Author

Reviewer #1: 1. The images in the text did not display properly. Please ensure that there are no formatting errors in the final submitted version

Table 1 should have 3 columns, each column consisting of parameter name, parameter meaning, and parameter value

In the "Steering control system optimization design", a Linear 2-degree-of-freedom model was adopted, which means that this article only considers sideslip and yaw at the theoretical level, without considering roll and pitch. But the rollover caused by vehicle roll is also a serious problem. Is the dynamic model used in this article reasonable?

4. It is recommended to conduct tests under classic serpentine and hook conditions to verify the effectiveness of the proposed method in this paper

5. For experimental results, in addition to visualized curves, it is recommended to provide quantifiable indicator results.

Why is PID used as the optimization controller in the article, and what are the advantages of this method compared to other methods?

7. This article uses existing dynamic models and control methods, and the innovation of this article is not clear enough

8. The Introduction section lacks a comprehensive summary of existing research. It is recommended to cite "A review on reinforcement learning based highway autonomous vehicle control" and "Longitudinal and lateral control methods from single vehicle to autonomous platform"

Reviewer #2: The manuscript investigates the steering stability of the XZJ5240JQZ30 all-terrain rescue lift vehicle under Double Line Change conditions. It develops a state-feedback PID optimization control system based on a two-degree-of-freedom vehicle model to enhance the vehicle's stability during steering maneuvers. Simulation results demonstrate that the proposed control system effectively improves the vehicle's stability and steering performance at both high and low speeds.

Below are my comments and suggestions.

1. Methodological aspects

1) Model Assumptions: The linear two-degree-of-freedom (2DOF) model is used for vehicle stability analysis. While this model is a good starting point, it may not fully capture the complex dynamics of a heavy rescue vehicle, especially under extreme conditions. Suggestion: Consider incorporating a more sophisticated model, such as a nonlinear model or a 3DOF model, to better represent the vehicle's behavior under various steering and speed scenarios.

2) Parameter Identification: The paper does not detail how the parameters for the mathematical models were identified or validated. Suggestion: Include a section on parameter identification, describing the methods used to determine the values of key parameters like suspension stiffness and damping coefficients. This could involve experimental data fitting or sensitivity analysis to ensure the model's accuracy.

3) Simulation Validation: The study relies heavily on simulations, but there is limited discussion on how these simulations were validated against real-world data. Suggestion: Conduct experimental tests with the actual XZJ5240JQZ30 vehicle or similar models to validate the simulation results. This would enhance the credibility of the findings and demonstrate the practical applicability of the proposed control system.

2. Technical aspects

1) Control System Design: The PID control system is used for optimization, but the choice of PID gains (P=8, I=5, D=1) seems arbitrary. Suggestion: Explain the process of tuning these gains, possibly using a systematic approach like Ziegler-Nichols method or model-based optimization techniques. This would provide a rationale for the chosen values and ensure optimal performance.

2) Handling of Nonlinearities: The control system may face challenges in handling the nonlinearities inherent in vehicle dynamics, such as tire force saturation or suspension nonlinearities. Suggestion: Explore the use of advanced control techniques like sliding mode control or model predictive control, which are better suited for nonlinear systems. These methods can provide robustness and improved performance under varying conditions.

3) Real-time Implementation: The study focuses on simulation results, but there is no discussion on the real-time implementation of the control system. Suggestion: Address the computational requirements and feasibility of implementing the control algorithm in real-time on the vehicle's onboard systems. Consider factors like processing power, response time, and sensor data availability.

4) Robustness to Disturbances: The impact of external disturbances, such as road irregularities or wind gusts, on the vehicle's stability is not explicitly considered. Suggestion: Incorporate disturbance rejection techniques in the control system design, such as using disturbance observers or robust control strategies, to ensure stability and performance in the presence of real-world uncertainties.

Below are my comments and suggestions to improve the writing and presentation of the manuscript.

1. Title and Abstract

Title: The title is quite lengthy. Consider simplifying it to highlight the core content and innovation of the study. For example, change "Study on optimization control of rescue hoist vehicle safety performance based on state feedback: A research on the XZJ5240JQZ30 All-Terrain Rescue Lift Vehicle" to "Optimization Control of All-Terrain Rescue Lift Vehicle Safety Performance Based on State Feedback".

Abstract: The abstract lacks a brief description of the research methods. It is suggested to add a sentence about how the 2DOF vehicle model and state-feedback PID optimization control system were established.

2. Introduction

Background Context: The introduction could benefit from a clearer linkage between the background information and the specific objectives of this study. Consider adding a paragraph that explicitly states how the existing challenges in rescue vehicle performance motivated this research.

Literature Review: While several studies are cited, there is room for a more comprehensive review. Include a broader range of relevant literature to establish the gap your study aims to fill. For instance, discuss more extensively how previous research has addressed the stability and steering control of heavy vehicles in off-road conditions.

3. Methodology

Mathematical Models: The mathematical models and equations are central to the study. Ensure that all symbols and variables are clearly defined and consistently used throughout the text. Consider providing a table summarizing the key parameters and their units for easy reference.

Simulation Setup: The description of the simulation experiments could be more detailed. Specify the exact parameters and conditions used in the TruckSim and Simulink simulations, such as the exact specifications of the road surface and environmental conditions. This will enhance the reproducibility of the study.

4. Results

Data Presentation: While the results are presented, they could be more effectively visualized. Consider using additional graphs or charts to compare the performance metrics of the 4WS and FWS systems more clearly. For example, a side-by-side comparison of yaw rate and lateral offset for both systems at different speeds would be helpful.

Statistical Analysis: Include statistical analysis to quantify the significance of the differences observed between the 4WS and FWS systems. This will strengthen the validity of your findings.

5. Discussion

Interpretation of Results: The discussion section should more thoroughly interpret the results in the context of the research objectives. Explain how the findings contribute to the understanding of rescue vehicle stability and steering control, and how they can be applied in practical scenarios.

Limitations: Address the limitations of the study more explicitly. Discuss any assumptions made in the model or simulations that might affect the results, and suggest areas for future research to address these limitations.

6. Conclusion

Summary of Findings: The conclusion should succinctly summarize the key findings and their implications. Clearly state how the state-feedback PID optimization control system enhances the stability of the rescue lift vehicle.

Future Work: Suggest potential directions for future research. For example, consider how the control system could be adapted for different types of rescue vehicles or tested under more varied environmental conditions.

7. Writing and Language

Clarity and Conciseness: Some sections contain lengthy sentences that could be broken down for clarity. Aim for concise and straightforward language to improve readability.

Technical Jargon: Ensure that technical terms and jargon are explained or defined for readers who may not be experts in the field. This will make the paper more accessible to a broader audience.

8. Figures and Tables

Figure Quality: Ensure that all figures are of high quality and clearly labeled. The captions should provide sufficient information to understand the figures without referring to the text.

Table Organization: If tables are used, ensure they are well-organized and easy to read. Use consistent formatting and include clear headings and units.

By addressing these comments and suggestions, the manuscript can be strengthened and made more suitable for publication.

6. PLOS authors have the option to publish the peer review history of their article (what does this mean? ). If published, this will include your full peer review and any attached files.

**Do you want your identity to be public for this peer review?** For information about this choice, including consent withdrawal, please see our Privacy Policy .

Reviewer #1: No

Reviewer #2: No

---

## [Author Response · Author response to Decision Letter 0]

15 Mar 2025

Dear PhD. Zhang,

Thanks for your letter and for reviewers’ comments concerning our manuscript entitled “Optimization control of all-terrain rescue lift vehicle safety performance based on state feedback” (Manuscript Number: PONE-D-24-59867). Those comments are all valuable and helpful for revising and improving our paper. We have studied all comments carefully and have made conscientious correction. During the revision process, we have also made adjustments to the manuscript format and related documents to ensure compliance with PLOS ONE's style requirements. Additionally, regarding the discrepancy in the funding information provided in the "Funding Information" and "Financial Disclosure" sections that you pointed out, we have corrected the relevant details in our submission. The main corrections in the paper and the responds to the reviewers’ comments are as flowing.

Reviewer Comments & Author Responses

Reviewer Reports on the Initial Version:

Reviewer # 1

1. The images in the text did not display properly. Please ensure that there are no formatting errors in the final submitted version.

2. Table 1 should have 3 columns, each column consisting of parameter name, parameter meaning, and parameter value.

3. In the "Steering control system optimization design", a Linear 2-degree-of-freedom model was adopted, which means that this article only considers sideslip and yaw at the theoretical level, without considering roll and pitch. But the rollover caused by vehicle roll is also a serious problem. Is the dynamic model used in this article reasonable?

4. It is recommended to conduct tests under classic serpentine and hook conditions to verify the effectiveness of the proposed method in this paper.

5. For experimental results, in addition to visualized curves, it is recommended to provide quantifiable indicator results.

6. Why is PID used as the optimization controller in the article, and what are the advantages of this method compared to other methods?

7. This article uses existing dynamic models and control methods, and the innovation of this article is not clear enough.

8. The Introduction section lacks a comprehensive summary of existing research. It is recommended to cite "A review on reinforcement learning based highway autonomous vehicle control" and "Longitudinal and lateral control methods from single vehicle to autonomous platform".

Reviewer # 2

The manuscript investigates the steering stability of the XZJ5240JQZ30 all-terrain rescue lift vehicle under Double Line Change conditions. It develops a state-feedback PID optimization control system based on a two-degree-of-freedom vehicle model to enhance the vehicle's stability during steering maneuvers. Simulation results demonstrate that the proposed control system effectively improves the vehicle's stability and steering performance at both high and low speeds.

Below are my comments and suggestions.

1. Methodological aspects

1) Model Assumptions: The linear two-degree-of-freedom (2DOF) model is used for vehicle stability analysis. While this model is a good starting point, it may not fully capture the complex dynamics of a heavy rescue vehicle, especially under extreme conditions. Suggestion: Consider incorporating a more sophisticated model, such as a nonlinear model or a 3DOF model, to better represent the vehicle's behavior under various steering and speed scenarios.

2) Parameter Identification: The paper does not detail how the parameters for the mathematical models were identified or validated. Suggestion: Include a section on parameter identification, describing the methods used to determine the values of key parameters like suspension stiffness and damping coefficients. This could involve experimental data fitting or sensitivity analysis to ensure the model's accuracy.

3) Simulation Validation: The study relies heavily on simulations, but there is limited discussion on how these simulations were validated against real-world data. Suggestion: Conduct experimental tests with the actual XZJ5240JQZ30 vehicle or similar models to validate the simulation results. This would enhance the credibility of the findings and demonstrate the practical applicability of the proposed control system.

2. Technical aspects

1) Control System Design: The PID control system is used for optimization, but the choice of PID gains (P=8, I=5, D=1) seems arbitrary. Suggestion: Explain the process of tuning these gains, possibly using a systematic approach like Ziegler-Nichols method or model-based optimization techniques. This would provide a rationale for the chosen values and ensure optimal performance.

2) Handling of Nonlinearities: The control system may face challenges in handling the nonlinearities inherent in vehicle dynamics, such as tire force saturation or suspension nonlinearities. Suggestion: Explore the use of advanced control techniques like sliding mode control or model predictive control, which are better suited for nonlinear systems. These methods can provide robustness and improved performance under varying conditions.

3) Real-time Implementation: The study focuses on simulation results, but there is no discussion on the real-time implementation of the control system. Suggestion: Address the computational requirements and feasibility of implementing the control algorithm in real-time on the vehicle's onboard systems. Consider factors like processing power, response time, and sensor data availability.

4) Robustness to Disturbances: The impact of external disturbances, such as road irregularities or wind gusts, on the vehicle's stability is not explicitly considered. Suggestion: Incorporate disturbance rejection techniques in the control system design, such as using disturbance observers or robust control strategies, to ensure stability and performance in the presence of real-world uncertainties.

Below are my comments and suggestions to improve the writing and presentation of the manuscript.

1. Title and Abstract

Title: The title is quite lengthy. Consider simplifying it to highlight the core content and innovation of the study. For example, change "Study on optimization control of rescue hoist vehicle safety performance based on state feedback: A research on the XZJ5240JQZ30 All-Terrain Rescue Lift Vehicle" to "Optimization Control of All-Terrain Rescue Lift Vehicle Safety Performance Based on State Feedback".

Abstract: The abstract lacks a brief description of the research methods. It is suggested to add a sentence about how the 2DOF vehicle model and state-feedback PID optimization control system were established.

2. Introduction

Background Context: The introduction could benefit from a clearer linkage between the background information and the specific objectives of this study. Consider adding a paragraph that explicitly states how the existing challenges in rescue vehicle performance motivated this research.

Literature Review: While several studies are cited, there is room for a more comprehensive review. Include a broader range of relevant literature to establish the gap your study aims to fill. For instance, discuss more extensively how previous research has addressed the stability and steering control of heavy vehicles in off-road conditions.

3. Methodology

Mathematical Models: The mathematical models and equations are central to the study. Ensure that all symbols and variables are clearly defined and consistently used throughout the text. Consider providing a table summarizing the key parameters and their units for easy reference.

Simulation Setup: The description of the simulation experiments could be more detailed. Specify the exact parameters and conditions used in the TruckSim and Simulink simulations, such as the exact specifications of the road surface and environmental conditions. This will enhance the reproducibility of the study.

4. Results

Data Presentation: While the results are presented, they could be more effectively visualized. Consider using additional graphs or charts to compare the performance metrics of the 4WS and FWS systems more clearly. For example, a side-by-side comparison of yaw rate and lateral offset for both systems at different speeds would be helpful.

Statistical Analysis: Include statistical analysis to quantify the significance of the differences observed between the 4WS and FWS systems. This will strengthen the validity of your findings.

5. Discussion

Interpretation of Results: The discussion section should more thoroughly interpret the results in the context of the research objectives. Explain how the findings contribute to the understanding of rescue vehicle stability and steering control, and how they can be applied in practical scenarios.

Limitations: Address the limitations of the study more explicitly. Discuss any assumptions made in the model or simulations that might affect the results, and suggest areas for future research to address these limitations.

6. Conclusion

Summary of Findings: The conclusion should succinctly summarize the key findings and their implications. Clearly state how the state-feedback PID optimization control system enhances the stability of the rescue lift vehicle.

Future Work: Suggest potential directions for future research. For example, consider how the control system could be adapted for different types of rescue vehicles or tested under more varied environmental conditions.

7. Writing and Language

Clarity and Conciseness: Some sections contain lengthy sentences that could be broken down for clarity. Aim for concise and straightforward language to improve readability.

Technical Jargon: Ensure that technical terms and jargon are explained or defined for readers who may not be experts in the field. This will make the paper more accessible to a broader audience.

8. Figures and Tables

Figure Quality: Ensure that all figures are of high quality and clearly labeled. The captions should provide sufficient information to understand the figures without referring to the text.

Table Organization: If tables are used, ensure they are well-organized and easy to read. Use consistent formatting and include clear headings and units.

By addressing these comments and suggestions, the manuscript can be strengthened and made more suitable for publication.

Author Responses to Initial Comments:

Reviewer # 1 (Remarks to the Author):

Reply: We sincerely appreciate your meticulous review of our manuscript and the constructive feedback you have provided. We have provided detailed responses to the issues you raised and have made revisions to address the shortcomings in the manuscript. We look forward to your review of our revised submission and your valuable feedback.

1. The images in the text did not display properly. Please ensure that there are no formatting errors in the final submitted version.

Reply: We sincerely apologize for the inconvenience caused by our oversight, which affected your reading experience. During the revision process, we have adjusted all the images included in the manuscript. We hope these modifications will enhance your reading experience.

For this comment, the relevant changes are shown in the new version of the figure.

2. Table 1 should have 3 columns, each column consisting of parameter name, parameter meaning, and parameter value.

Reply: During the revision process, we have added the definitions of various parameters in Table 1. These modifications will help readers better understand each parameter.

For this comment, the revised manuscript is shown below, with the modified sections marked in blue.

Table 1 Basic Parameters of the All-Terrain Rescue Lift Vehicle

Parameter Name Parameter meaning Parameter Value

Vehicle mass Total vehicle weight 24000kg

Vehicle length Total length of the vehicle 11575mm

Vehicle width Total width of the vehicle 2500mm

Vehicle height Total height of the vehicle 3855mm

Wheel base Distance between front and rear axles 3505mm

Distance from CG to front wheel axle Horizontal distance from CG to front axle 1935mm

Center of mass height Vertical height of the center of mass 1650mm

Tire specification Tire model and load capacity 385/95R25 170F

3. In the "Steering control system optimization design", a Linear 2-degree-of-freedom model was adopted, which means that this article only considers sideslip and yaw at the theoretical level, without considering roll and pitch. But the rollover caused by vehicle roll is also a serious problem. Is the dynamic model used in this article reasonable?

Reply: Thank you for your highly constructive comments. The selection of the linear 2DOF model in this study was based on actual vehicle parameters and accident occurrence rates. Given that the all-terrain rescue lift vehicle has a relatively low center of mass and a large weight, sideslip and yaw are the more common accident phenomena, whereas reports of roll and pitch accidents are relatively rare. Additionally, data obtained from a mining group in China support this observation. Therefore, while we acknowledge that, as you mentioned, vehicle rollover due to tilting is also a serious issue, from the perspective of addressing practical engineering problems, it is reasonable to focus solely on the analysis and optimization of sideslip and yaw phenomena.

For this comment, no modifications were necessary in the relevant part of the manuscript.

4. It is recommended to conduct tests under classic serpentine and hook conditions to verify the effectiveness of the proposed method in this paper.

Reply: Thank you for your valuable comments. The simulation conditions in this study were designed based on the actual conditions of mining roads.Since the vehicles operating on mining roads are mostly large and heavy mining engineering vehicles, the road design, based on their intended use and the actual conditions of the mining site, does not involve serpentine conditions with continuous turns. Instead, the planned turning routes are relatively smooth. However, even under such conditions, these large and heavy mining vehicles still experience sideslip and yaw during turning. This study was conducted based on such an engineering application scenario.

For this comment, no modifications were necessary in the relevant part of the manuscript.

5. For experimental results, in addition to visualized curves, it is recommended to provide quantifiable indicator results.

Reply: To improve the presentation of the experimental results, we have optimized the images to provide a clearer comparison of the results. Additionally, to facilitate quantitative analysis, we have created a table displaying the specific peak values of the curves under both 4WS and FWS conditions. This allows for a clearer demonstration of the effectiveness of the proposed PID control method in enhancing the stability of the all-terrain rescue lift vehicle.

For this comment, the revised manuscript is shown below, with the modified sections marked in blue.

4.3. Analysis of vehicle stability at high speeds

In emergency situations, the rescue team needs to drive the all-terrain rescue lift vehicle at high speeds while ensuring both safety and maneuverability. Therefore, in this simulation experiment, the vehicle speed was set to 70 km/h to compare and analyze the performance parameters of the four-wheel steering (4WS) all-terrain rescue lift vehicle with PID control and the FWS all-terrain rescue lift vehicle without PID control. The results are shown in Figure 12.

Fig. 12. Parameter curve changes at high speed

As illustrated in Figure 12(a), during high-speed driving, the trajectory of the 4WS vehicle with PID control exhibits a slight deviation compared to the FWS vehicle without PID control from the initial steering phase until 3.2 seconds. However, from 3.2 seconds until the conclusion of the simulation, the 4WS vehicle with PID control closely adheres to the intended path. In high-speed driving scenarios, PID control demonstrates a substantial stabilizing effect on the all-terrain rescue lift vehicle, particularly on curved road sections. To conduct a quantitative analysis of the simulation results, the peak values of the steering wheel angle, vehicle slip angle, and vehicle yaw angle were extracted and are presented in Table 3.

Table 3 Peak data at high speed

Variable Name 4WS FWS

Steerin

---

## [Decision Letter · Decision Letter 1]

7 Apr 2025

Optimization control of all-terrain rescue lift vehicle safety performance based on state feedback

PONE-D-24-59867R1

Dear Dr. Tian,

We’re pleased to inform you that your manuscript has been judged scientifically suitable for publication and will be formally accepted for publication once it meets all outstanding technical requirements.

Kind regards,

Lei Zhang, PhD

Academic Editor

PLOS ONE

Additional Editor Comments (optional):

The revised version has addressed all ther raised comments, and thus can be accepted for publication.

Reviewers' comments:

Reviewer's Responses to Questions

**Comments to the Author**

1. If the authors have adequately addressed your comments raised in a previous round of review and you feel that this manuscript is now acceptable for publication, you may indicate that here to bypass the “Comments to the Author” section, enter your conflict of interest statement in the “Confidential to Editor” section, and submit your "Accept" recommendation.

Reviewer #1: (No Response)

Reviewer #3: All comments have been addressed

2. Is the manuscript technically sound, and do the data support the conclusions?

Reviewer #1: (No Response)

Reviewer #3: Yes

3. Has the statistical analysis been performed appropriately and rigorously? 

Reviewer #1: (No Response)

Reviewer #3: Yes

4. Have the authors made all data underlying the findings in their manuscript fully available?

Reviewer #1: (No Response)

Reviewer #3: Yes

5. Is the manuscript presented in an intelligible fashion and written in standard English?

Reviewer #1: (No Response)

Reviewer #3: Yes

6. Review Comments to the Author

Reviewer #1: (No Response)

Reviewer #3: The article adopts a linear 2-degree-of-freedom (2DOF) model, which only considers yaw and sideslip, ignoring roll and pitch dynamics. For heavy rescue vehicles, roll stability is equally critical (such as rollover risk). It is recommended to supplement the analysis with a 3-degree-of-freedom model or a nonlinear model, or clarify the limitations of this model in the discussion section.

7. PLOS authors have the option to publish the peer review history of their article (what does this mean? ). If published, this will include your full peer review and any attached files.

**Do you want your identity to be public for this peer review?** For information about this choice, including consent withdrawal, please see our Privacy Policy .

Reviewer #1: No

Reviewer #3: No

---

## [Editor Report · Acceptance letter]

PONE-D-24-59867R1

PLOS ONE

Dear Dr. Tian,

I'm pleased to inform you that your manuscript has been deemed suitable for publication in PLOS ONE. Congratulations! Your manuscript is now being handed over to our production team.

Kind regards,

on behalf of

Dr. Lei Zhang

Academic Editor

PLOS ONE